# Optimal spectral templates for triggered feedback experiments

**Anand S. Kulkarni**[1,2], **Todd W. Troyer**[1,2]*

**1** Biology Department, University of Texas at San Antonio, San Antonio, Texas, United States America,
**2** UTSA Neurosciences Institute, University of Texas at San Antonio, San Antonio, Texas, United States America

* todd.troyer@utsa.edu

## Abstract

In the field of songbird neuroscience, researchers have used playback of aversive noise bursts to drive changes in song behavior for specific syllables within a bird's song. Typically, a short (~5–10 msec) slice of the syllable is selected for targeting and the average spectrum of the slice is used as a template. Sounds that are sufficiently close to the template are considered a match. If other syllables have portions that are spectrally similar to the target, false positive errors will weaken the operant contingency. We present a gradient descent method for template optimization that increases the separation in distance between target and distractors slices, greatly improving targeting accuracy. Applied to songs from five adult Bengalese finches, the fractional reduction in errors for sub-syllabic slices was 51.54±22.92%. At the level of song syllables, we use an error metric that controls for the vastly greater number of distractors vs. target syllables. Setting 5% average error (misses + false positives) as a minimal performance criterion, the number of targetable syllables increased from 3 to 16 out of 61 syllables. At 10% error, targetable syllables increased from 11 to 26. By using simple and robust linear discriminant methods, the algorithm reaches near asymptotic performance when using 10 songs as training data, and the error increases by <2.3% on average when using only a single song for training. Targeting is temporally precise, with average jitter of 3.33 msec for the 16 accurately targeted syllables. Because the algorithm is concerned only with the problem of template selection, it can be used as a simple and robust front end for existing hardware and software implementations for triggered feedback.

## Introduction

For over a century, operant conditioning has had great success in making complex learning questions experimentally tractable. Combined with electrophysiological recording, it has been the main experimental tool for understanding the neural underpinnings of learning processes [1, 2]. Applying operant conditioning to discrete behaviors such as lever presses and nose pokes is relatively straightforward. However, to successfully shape a complex behavior, a conditioning stimulus must be delivered with high specificity and with a short temporal delay [3].

In the past decade, short-delay operant feedback techniques have been applied fruitfully in the field of birdsong neuroscience. An individual bird's song comprises a sequence of distinct

**Funding:** This work was supported by National Science Foundation (https://www.nsf.gov/) grants DBI-1451032 and IOS-0951310 to TWT. The funder had no role in study design, data collection and analysis, decision to publish, or preparation of the manuscript.

**Competing interests:** The authors have declared that no competing interests exist.

vocal elements, called syllables, learned by a process of imitation early in life [4]. Operant conditioning experiments generally involve real-time detection of a targeted portion within the song, followed by playback of a burst of aversive white noise if a specific acoustic parameter falls on one side of a preset trigger threshold. This technique, which we term Triggered Feedback (TF), has been used to drive changes in the pitch of specific portions of song [3, 5–8], alter the duration of individual song syllables [9], and to drive changes in syllable transition probabilities in birds with variable sequencing [10, 11]. It has given investigators a granular ability to induce learning and examine the relevant behavioral and neural dynamics [12–20]. The TF approach has been paired with optogenetic stimulation within specific brain nuclei to delineate their roles in different aspects of song learning and production [21–24].

In typical experiments in the birdsong field using TF [7–12, 14–16, 21–24], an individual syllable is selected as a target for real-time detection based on manual inspection of song spectrograms (*e.g.* syllable 'G,' Fig 1A and 1D). Candidate syllables generally have a prominent and unique harmonic structure that remains stable for a significant portion of the syllable. A spectral template is formed by averaging this portion across several instances of the syllable (Fig 1D–1F). During the experiment, the digital audio stream is segmented into temporal slices and a normalized spectrum obtained from each slice in real time (Fig 1B; see Methods). If the distance between this spectrum and the template is less than a pre-set threshold, it is considered a

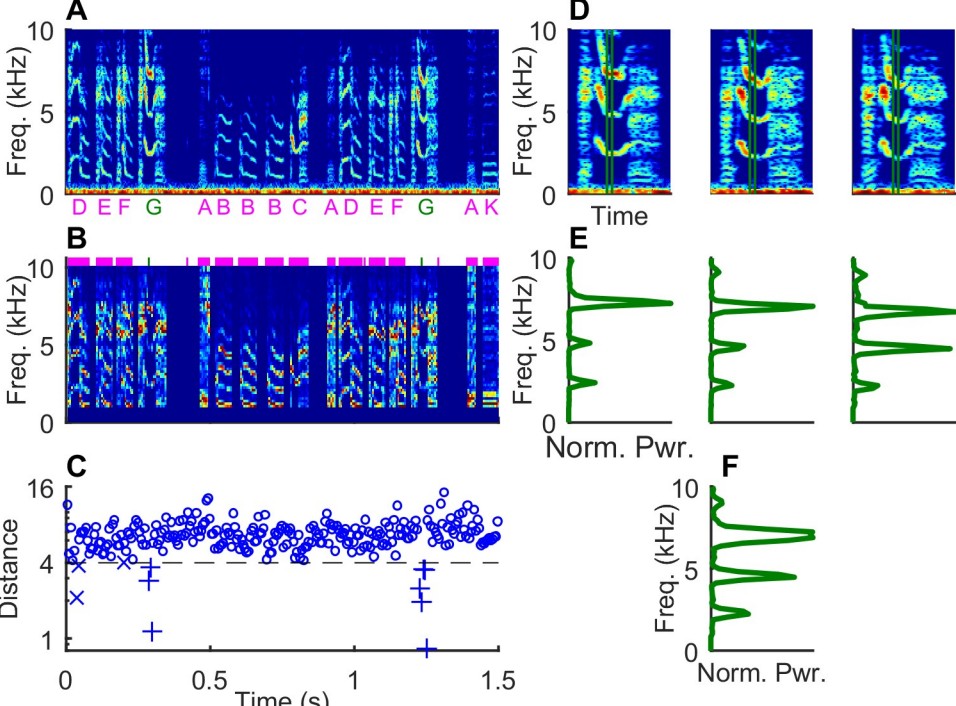

**Fig 1. Matching an averaged template** (A) Spectrogram of 1.5 sec of Bengalese finch song with syllable labels shown below. 'G' will be used as an example target syllable throughout the paper. (B) Normalized spectrogram corresponding to A. Slices are ~6ms long and non-overlapping. Frequencies less than 1000 Hz are set to zero and remaining amplitudes are linearly scaled per slice to fit in the range 0–1. Green ticks: target slices at a fixed position within syllable 'G.' Magenta bars: distractor slices. For visual clarity, inter-syllable gaps are not shown. (C) Euclidian distance of the spectral slices to a template made by averaging all target slices. '+' and 'x' symbols indicate slices with distances less than the threshold (dotted line) belonging to target and distractor categories respectively. (D) Spectrograms of three instances of the target syllable 'G' with the target slice (7th slice out of 16) highlighted by green lines. (E) Normalized spectra of the target slice from the three instances. (F) Normalized average of the three spectra shown in E.

'match' ('+' symbols, Fig 1C). If the match criterion is satisfied by a set number of consecutive slices, a 'detection' occurs. If other syllables in the bird's repertoire have portions that are similar ('x' symbols, Fig 1C) to the template, the resulting errors in detection weaken the operant contingency and reduce the effectiveness of the experiment.

In this paper, we present a systematic method for template construction that minimizes errors in detection. After smoothing the distribution of distances to the template, we use gradient descent within the space of spectral vectors to derive an optimized template that minimizes detection errors. Since our algorithm works to minimize both false positive and false negative errors, the optimized template depends on the spectral structure of distractors as well as that of the target syllable. As a result, even small spectral differences between the target syllable and distractors can be used to significantly increase detection accuracy. We tested our algorithm by measuring detection performance for 61 syllables sung by five adult Bengalese finches. Using averaged templates, only 3 (4.9%) syllables were suitable experimental targets based on their detection accuracy (<5% average error). Our algorithm improved detection accuracy significantly and increased the number of potential experimental targets to 16 syllables (26.2%).

## Methods

### Data acquisition and pre-processing

Amplitude-based triggering was used to record and digitize data from five Bengalese finches singing alone ("undirected song") in a sound isolation chamber. Two different recording systems were used: EvTAF [3](44150 Hz sampling rate, three birds) and Sound Analysis Pro (44100 Hz, two birds)[25]. Songs were defined as continuous periods of singing separated by at least two (three birds) or three (two birds) seconds of silence. For each bird, fifty songs were randomly selected from a single day. Twenty-five songs were used as a training set for the optimization process and to assess classification performance at the level of individual spectral slices. The other twenty-five were used as a test set to assess syllable-level classification.

All procedures were performed in accordance with established animal care protocols approved by the University of Texas at San Antonio's Institutional Animal Care and Use Committee (IACUC; Protocol # IS0058).

### Song annotation

An offline procedure was used to segment and annotate song files. For segmentation, the raw audio signal was first filtered using a zero-phase, bandpass filter between 1000 and 10,000 Hz. The signal was then squared and convolved with a two msec square window to determine the amplitude envelope. Songs were segmented by manual optimization of three thresholds (set separately for each bird). First, an amplitude threshold divided the song into segments with high amplitude. Second, short-duration sub-threshold gaps between high amplitude segments were eliminated and the adjacent high amplitude segments were merged. Finally, any high amplitude segments shorter than a threshold duration were eliminated. The minimum duration for gap elimination averaged 5.4 msec across birds (range 2–10 msec). The minimum duration for segments was 10 (4 birds) or 15 (1 bird) msec. This resulted in a data set containing 30,229 segments (4477–8894 per bird).

For each bird, we classified the segments into distinct categories based on manual inspection of their spectral structure and the context in which they occurred. Categories were organized into two types: stereotyped vocalizations sung as part of the song (syllables) and other sounds. The latter included three sub-categories: heterogeneous vocalizations made outside the context of the song (calls etc.), non-vocal noises (wing flaps, cage noises etc.), and short "clicks" (median length <15 msec, [26]). Across five birds, there were 76 categories (61

syllables and 15 other categories; range 10–23 per bird; median 15). Syllables constituted 23,819 of the 29,743 segments (80.11%), with 3548 out of the 5915 non-syllable segments (59.98%) due to calls and noise produced by a single bird before or after the main song sequence. Due to occasional erroneous segmentation, some category instances were broken down into multiple segments while other category instances were combined with neighbors to form composite super-segments. These instances were excluded from our optimization algorithm (495 segments; 1.64% overall and 0.50% to 3.06% of segments per bird).

The audio, segmentation, and labeling data for the five birds used here has been deposited online for replication and further use (Harvard Dataverse, doi:10.7910/DVN/62EHNM)

## Normalizing and averaging spectral slices

All analysis was performed using custom scripts written in MATLAB (Natick, MA). These have been uploaded to GitHub as a public repository along with relevant documentation (https://github.com/toddtroyerlab/template_optimization).

All calculations were performed offline, but the template-based detection is designed to mimic the real-time TF algorithm described in [3]. Other TF methods are essentially similar. We divided the raw audio files into non-overlapping slices that were 256 samples (~6 msec) long. Each slice was transformed by subtracting the mean, multiplying by a hamming window, and using a fast Fourier transform to calculate the spectrum. Spectral amplitudes less than 1000 Hz were set to zero. The remaining amplitudes were normalized between 0 and 1 by linear interpolation between the minimum and maximum amplitudes (Fig 1B). Henceforth the word slice will refer to this normalized spectrum. All slices were assigned a label based on the location of center of the slice. Slices whose center fell between segments were labeled as 'gap' slices.

For each slice, song amplitude was calculated as the sum of the squared slice spectrum (before normalization). An amplitude threshold was set for each bird by finding the amplitude that yields the optimal discrimination between the amplitude distribution of syllable slices and gap slices within the training data from that bird. Only those gap slices that had higher amplitude than the threshold were used in the optimization algorithm as distractors.

To create a template based on averaged spectra, one must select spectral slices at the same position across several instances of a given syllable within the training set. But, due to natural variation in duration, different instances can contain different number of slices. First, we calculated the mean and standard deviation of syllable duration for each syllable category in our training set and discarded any instance greater than two standard deviations from the mean. We then aligned the remaining instances to the modal slice-length for that syllable using linear interpolation. For example, there were 206 instances of syllable G in the 25 training songs for bird 326. The mean duration was 93 msec with a standard deviation of 3 msec. Eight instances had durations less than 87 msec or greater than 99 msec and were excluded. Of the remaining 198, the number of instances with slice-lengths 15, 16, and 17 were 31, 121, and 46 respectively. All instances were aligned to have 16 slices using linear stretching of time. Finally, an averaged template was determined for each of the 16 slice positions by calculating a mean spectrum over all 198 aligned instances and normalizing the mean vector between 0 and 1. We used the training data to align and calculate templates for all 575 slice locations within the 61 syllables identified across 5 birds.

## Template optimization and evaluation

Optimized templates were calculated for each of the 575 slice locations within our training data using gradient descent (see next section). Each optimization required the specification of

an initial state and a set of target and distractor slices. The target slices consisted of the time aligned slices described above, and the initial condition was set to be the averaged template obtained by averaging these slices. Distractors included all slices assigned to categories other than the syllable of interest plus gap slices that exceeded the amplitude threshold.

Optimization used gradient descent with a variable step size determined using standard back-tracking line search [27]. The termination of gradient descent depended on: (A) the absolute value of gradient magnitude, (B) the change in gradient magnitude, and (C) the change in total error. B and C were calculated over the 10 most recent steps. Gradient descent was terminated when either C and B or C and A decreased below preset thresholds. If this did not happen for 1000 steps, the descent was terminated and data from the 1000th step was used for further processing. 67/575 (11.65%) optimizations were forced to terminate on the 1000th step. The median stopping step for the remaining optimizations was 261.

As described above, specifying an identified slice requires a process of temporal alignment. To avoid issues that arise from separate temporal alignments for the test and training set, we use target and distractor slices within the training data to evaluate performance of the algorithm at the level of song slices.

We then go on to measure targeting performance at the level of song syllables (see Results). The targeting performance of optimized templates is then evaluated on a separate set of test data using an algorithm that mimics the real-time TF algorithm presented in [3].

## Gradient descent

A typical method for TF is to construct a template for detection by selecting a portion of the song with a distinct spectral structure, and then averaging over multiple instances of that portion of the song. Such averaging implicitly assumes a method for temporal alignment across syllable instances. Therefore, to start, we used linear scaling of time to align all spectra to the modal slice length for each syllable category and calculated an averaged template for each slice position (see above). With this average as an initial template, we then systematically improve detection performance using gradient descent. In this section, we derive the gradient descent expression and offer an interpretation. To illustrate several aspects of our algorithm, we will use the 7th slice position (out of 16) from syllable G as a specific example (Fig 1D).

Given a template vector, one can easily calculate the Euclidian distance between the template and the target and distractor slices. For a given distance threshold, the false positive error rate is the fraction of distractor slices with distances less than or equal to the threshold. The false negative error rate is the fraction of target slices with distances above the threshold (Fig 2A). The total error is defined as the average of the false positive and false negative error rates.

To use gradient descent for template optimization, we need an error function that is a differentiable function of the template vector. Therefore, we convert the discrete collection of distances into a continuous distribution by smoothing with a Gaussian. Letting $G_\sigma(x)$ denote the Gaussian smoothing function, the smoothed distance distributions for targets and distractors ($T(x)$ and $D(x)$) can be written as:

$$T(x) = \frac{1}{M} \sum_i G_\sigma(x - d_i) \text{ and } D(x) = \frac{1}{N} \sum_j G_\sigma(x - d_j)$$

where we have assumed there are $M$ target slices with distances to the template $d_i$ and $N$ distractor slices with distances $d_j$. For a given value of the threshold ($\theta$), the continuous false negative error rate and false positive error rate are just the areas of these distributions that fall on

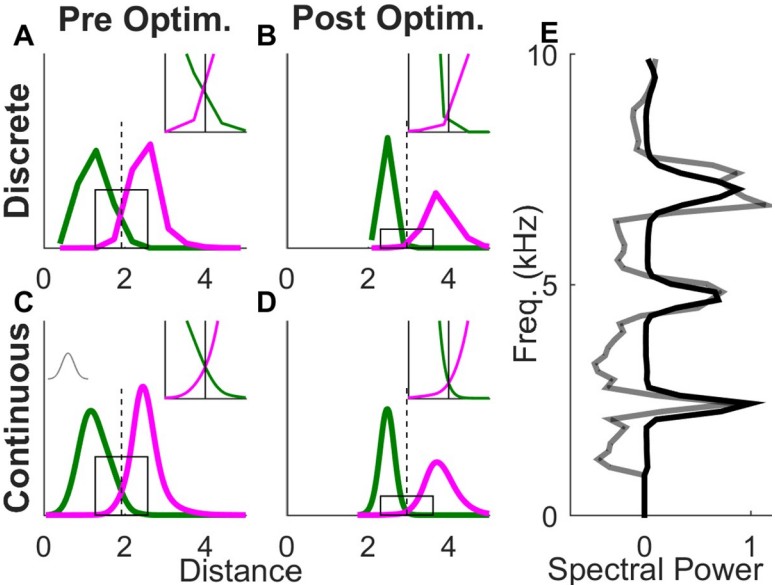

**Fig 2. Pre- and post-optimization distance distributions** (A-D) Probability density of target (green curve) and distractor (magenta curve) slice distances from the template for the 7$^{th}$ slice in example syllable G. The inset gives a magnified view of the distributions near threshold. (C & D) Smoothed distributions derived from A & C using a Gaussian smoothing function (grey curve inset in C). (A & C) Distance distributions using the average of target slices as the template (black curve in E). (C & D) Distance distributions using the optimized template (gray curve in E).

the "wrong" side of the threshold ([Fig 2C]). Thus, the total error (TE) is given by

$$\frac{1}{2M} \sum_i \int_\theta^\infty G_\sigma(x - d_i)dx + \frac{1}{2N} \sum_j \int_{-\infty}^\theta G_\sigma(x - d_j)dx$$

To calculate the gradient of the total error with respect to the template vector, we note that each target and distractor slice makes an independent contribution to the false negative or false positive rate. Let $t$ denote the template vector and $s_i$ denote a specific target vector whose distance to the template is given by $d_i$. Using the chain rule, the component of the gradient corresponding to $s_i$ is given by

$$\nabla_t(\int_\theta^\infty G_\sigma(x - d_i)dx) = \frac{\partial}{\partial d_i}(\int_\theta^\infty G_\sigma(x - d_i)dx) \times \nabla_t(d_i)$$

For the first term, we switch the order of the integral and then apply the fundamental theorem of calculus:

$$\frac{\partial}{\partial d_i}(\int_\theta^\infty G_\sigma(x - d_i)dx) = \int_\theta^\infty -G'_\sigma(x - d_i)dx = G_\sigma(\theta - d_i)$$

For the second term, we note that the gradient of the distance from the template to $s_i$ is equal to a unit vector pointing from the target vector to the template:

$$\nabla_t(d_i) = \frac{(t - s_i)}{d_i}$$

Thus, the contribution of target vector $s_i$ to the negative gradient is a vector pointing from the template toward $s_i$ whose magnitude is given by the Gaussian smoothing function applied

to the difference between the distance to the template and the threshold distance $\theta$:

$$-\nabla_t \left( \int_\theta^\infty G_\sigma(x - d_i)dx \right) = G_\sigma(\theta - d_i) \times \frac{(s_i - t)}{d_i}$$

The calculation for each distractor vector is similar and yields a vector that points in the opposite direction (from the distractor vector toward the template), again weighted by the Gaussian smoothing function. Thus, significant contributions to the gradient are confined to slices whose distance to the template is near threshold. This means that modification of the template vector during gradient descent maximally takes account of slices for which changes in the template will likely alter the side of the threshold they fall on. The gradient for the total error is simply the sum of the contributions for all the target and distractor slices:

$$-\nabla_t(TE) = \frac{1}{2M}\sum_i G_\sigma(\theta - d_i)\frac{(s_i - t)}{d_i} - \frac{1}{2N}\sum_j G_\sigma(\theta - d_j)\frac{(s_j - t)}{d_j}$$

The width of the Gaussian smoothing function (sigma) is chosen in a manner that ensures a unique crossing point (Fig 2C) between any given pair of distance distributions. We start with a default value (0.2) and iteratively change it in intervals of 0.05 until we find the smallest value that ensures that the smoothed target and distractor distributions decrease monotonically from their peak. During gradient descent, an initial value of sigma that works for the set of slice distances to the initial template is adopted. For each subsequent step in the gradient descent, the monotonicity condition is checked. If the distributions are found to be insufficiently smooth at any stage of the optimization, a new higher value of sigma is calculated (testing smoothness in increments of 0.05) and the entire optimization is restarted using this new sigma. We call the unique crossing point that results from the smoothing procedure the 'slice optimal' threshold. It is recalculated and used for error calculations at each step in the gradient descent.

The gradient calculation shows that discrimination performance is improved by moving the template vector toward target vectors and away from distractors with points closer to the decision surface making a greater contribution. By shifting the template away from the average of the target distribution while simultaneously adjusting the threshold distance, this procedure results in a decision surface that more accurately separates target and distractor vectors (S1 Fig).

## Results

### Slice level performance

The standard procedure for template construction for TF is to first select a target syllable containing a stable and distinct spectral structure, and then to average over multiple instances of that portion of the syllable. Our general strategy is to use the averaged template for a given position within a target syllable as our initial condition, and then use gradient descent to optimize detection performance. We evaluated the performance of the averaged and optimized templates by determining whether the distance to the template for target and distractor slices fell below the slice-optimal threshold determined from the smoothed distribution of distances for each template (see Methods; Fig 2).

We performed gradient-based template optimization for all 575 slice positions in 61 syllables across our five birds. For each slice position, we first identify the target and distractor slices from the training set (Fig 1B). All slices at the target position from the aligned instances of that syllable type are designated as target slices. Slices in non-target positions within the

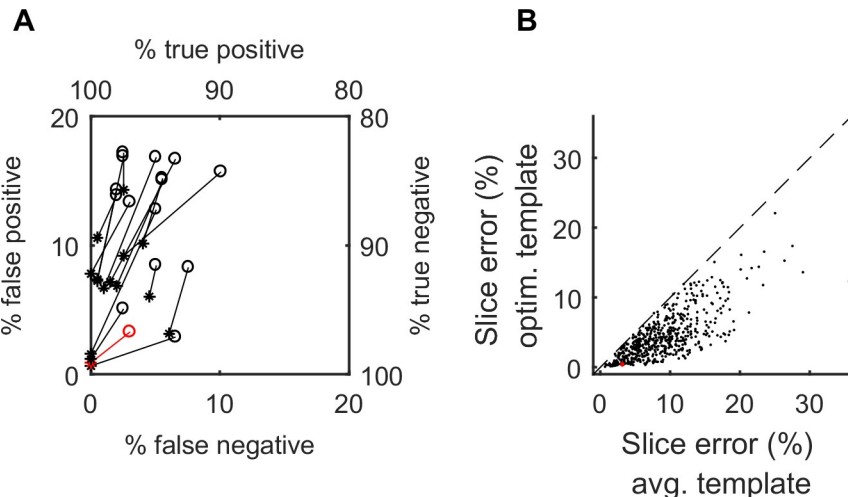

**Fig 3. Optimization improves slice classification** (A) False positive (% of distractor slices with sub-threshold distances) and false negative (% of target slices with supra-threshold distances) error rates for the 16 slice positions in the example syllable G. Performance for averaged (circles) and optimized (asterisks) templates for the same slice are connected by lines. (B) Slice error (mean of the false negative and false positive errors) for the averaged vs. optimized template (ordinate) for all slice positions from all target syllables. Red: performance for the 7th slice from syllable G (see Figs 1 and 2).

target syllable do not participate in optimization. Slices from instances of other syllables, and those from instances of non-syllable categories and gaps that exceed an amplitude threshold (see Methods) are designated as distractors.

Fig 3A shows the increase in classification performance for the 16 slice positions of the example syllable G. There is a robust decrease in the error for both target and distractor slices. For the 7th slice position, the misclassification of target slices was reduced from 3% to 0% and that of distractor slices from 3.3% to 0.88% (Fig 3A, data shown in red; also insets in Fig 2A vs 2B). Averaging the two error percentages gives us a single number to measure the classification performance; the slice error. Across the 16 slices in syllable G the slice error for averaged templates was 8.45±2.76% (mean±sd) versus 3.95±2.39% for the optimized templates, a fractional reduction of 56.14±24.31%. All summary statistics here onward are in the format mean±sd unless noted otherwise.

The relationship between the slice error for averaged and optimized templates for all slices is shown in Fig 3B. Across all birds and their syllable types, the slice error was reduced from 8.78±4.77 to 4.33±3.24, a fractional reduction of 51.54±22.92%. Note that in 2/575 slices, the slice error for the optimized template was slightly higher than that for the averaged template (although the increase was small, 0.04 and 0.08%). This likely stems from the fact that our algorithm performs gradient descent on the continuous error rate function whereas each slice is unambiguously classified as a match or non-match when reporting performance. In cases where gradient descent does little to reduce the continuous error rate, the discrete error rate can be slightly greater for the optimized template.

## Syllable level performance

Thus far our analysis treats song as a sequence of independent spectral slices of roughly 6 msec duration. However, syllables are generally viewed as units of song behavior, and in many experiments the intent of TF is to target specific syllables. Therefore, we analyzed the performance of our algorithm at the syllable level, with a syllable or gap considered detected if any

slice within that song element falls below the distance threshold and above the amplitude threshold.

Our syllable-level evaluation metric also addresses a second difficulty that stems from the imbalance in the number of targets versus distractors. As an example, our test data set contains 197 instances of syllable G and 6352 distractor syllables and gaps. If the false positive and false negative error rates were both 5%, this would result in roughly 10 false negatives and 150 false positives. To achieve a better balance between the two error types, we calculate the false positive rate as the number of false positives divided by the number of target syllables, rather than the number of distractors. We define balanced error to be the average of the false negative and this redefined false positive rate.

Since detecting any slice within a syllable counts as a detection, minimizing the syllable-level error requires more stringent criteria for syllable detection. In particular, the need for greater stringency is critical to prevent the detection of distractors. One way to achieve this is to reduce the distance threshold that determines whether a slice is considered a match to the template. To demonstrate the impact of varying the threshold, we plotted the syllable classification performance of the 16 optimized templates from the example syllable G when the distance threshold is varied between 50% and 100% of the slice-optimal threshold (different for each slice) in steps of 10% (Fig 4A–4C). As expected, a small distance threshold leads to high numbers of false negatives (Fig 4A), while a large threshold leads to high numbers of false positives (Fig 4B). The tradeoff results in a minimum value for the balanced error in the region of 80–90% of the slice-optimal value (Fig 4C).

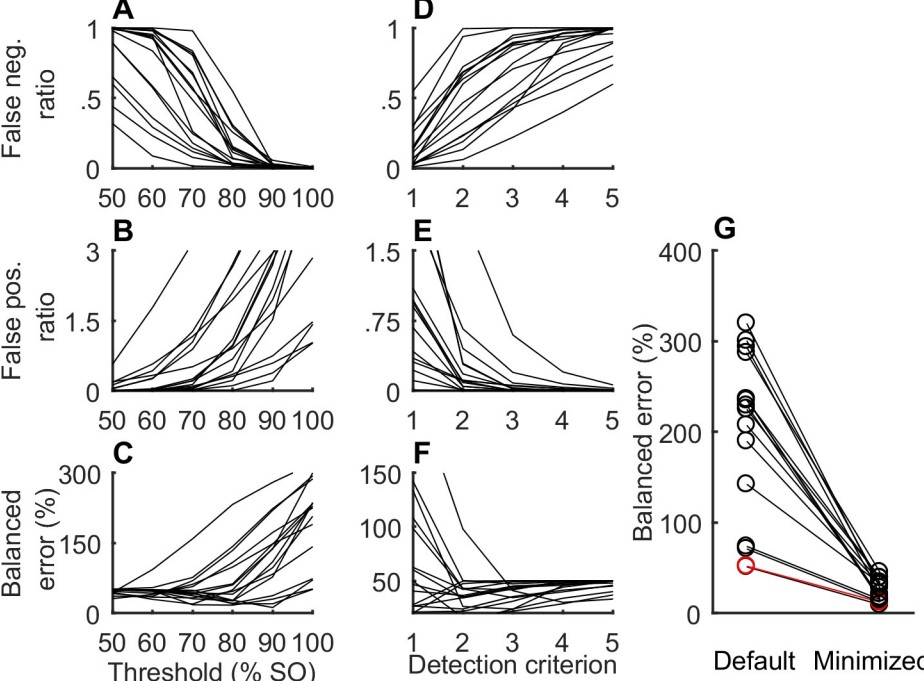

**Fig 4. Optimizing targeting parameters for syllable detection** (A) Ratio of false negative detections to the number of targets for different threshold values (given as percentages of the slice optimal threshold) for the 16 optimized templates from example syllable G (B) False positive ratio similar to A. (C) Balanced error (expressed as a percentage), equal to the average of false negative and false positive ratios shown in A and B (D-E) Same as A-C but for variations in the detection criterion (number of consecutive matches needed for a detection). Threshold level held constant at 80% of slice optimal threshold (G) Balanced error calculated using default targeting parameters (threshold level = 100% and detection criterion = 1) and the corresponding minimal values obtained by varying both targeting parameters. The example slice (7th) is shown in red.

Another method for increasing the stringency of syllable detection is to require that a syllable contain several consecutive sub-threshold slices to be detected. We plotted the classification performance for the 16 optimized templates from syllable G when the number of consecutive slices (termed the detection criterion) was varied from 1 to 5 (Fig 4D–4F) and the distance threshold held constant at 80%. For 12 of the 16 templates, the minimal values of the balanced error occurred for a detection criterion greater than one.

We used a combination of the two approaches to optimize syllable-level detection performance. For any given template, we calculated the balanced error for all (21x5) combinations of threshold (0% to 200% of slice-optimal threshold in steps of 10%) and detection criterion (1–5 consecutive matches) and chose the combination that gave minimal balanced error. For example, the balanced error for the 7$^{th}$ slice (red data, Fig 4G) with the threshold at 100% and detection criterion at 1 is 52%. On exploring all 21x5 combinations of threshold and detection criterion, the minimal balanced error of 12% is found at 80% threshold level and detection criterion of 2. For the 16 optimized templates in our example syllable G, optimizing the detection criterion and threshold resulted in a fractional reduction in balanced error of 85.44±5.8% over the default values of 1 (for detection criterion) and 100% of the slice optimal threshold (Fig 4G). Over the all 575 slices in our data set, the fractional reduction was 87.69±10.69%. The modal detection criterion was one slice; the median was 2 consecutive slices. The modal and mean value for threshold modification was a 10% reduction relative to the slice-optimal threshold.

## Syllable detection performance

Using the above strategy, we measured the minimal balanced error for the averaged and optimized templates for each of the 575 slice positions in our data set. Then, for each syllable we chose the averaged template with lowest balanced error (among the slice positions for that syllable). Similarly, we found the optimized template with lowest error for that syllable (Fig 5). Across the 61 syllables in our data set, template optimization resulted in a fractional reduction of 40.83±29.6% in the balanced error.

Setting 5% balanced error as acceptable performance, our method increased the yield of targetable syllables from 3 out of 61 for averaged templates (4.9%) to 16 out of 61 syllables after optimization (26.2%). Allowing a less stringent 10% error rate allows for the targeting of 11 of 61 syllables with averaged templates (18.0%) and 26 of 61 syllables with optimized templates (42.6%).

## Temporal precision

For experiments driving changes in a specific portion of song, TF must be presented with high temporal precision. To assess this, we measured timing jitter, defined as the standard deviation of the latency between syllable onset (based on the smoothed amplitude envelope; see Methods) and the onset of the slice where the syllable detection criteria were met. Since timing jitter can be affected by targeting accuracy, we focused our analysis on the 16 syllables that gave most accurate targeting. Since the latency to detection will be influenced by errors in the detection of syllable onsets resulting from extraneous cage noise or other factors, we excluded syllable instances that were outliers in the distribution of total syllable duration. Specifically, syllables whose duration exceeded the third quartile by more than 1.5 times the interquartile range (median 2.58% of syllable instances) were excluded. For the 16 accurately targetable syllables, timing was precise, with an average jitter between syllable onset and time of targeting of 3.33 ± 1.1 msec (mean ± std, range 1.93–5.59 msec).

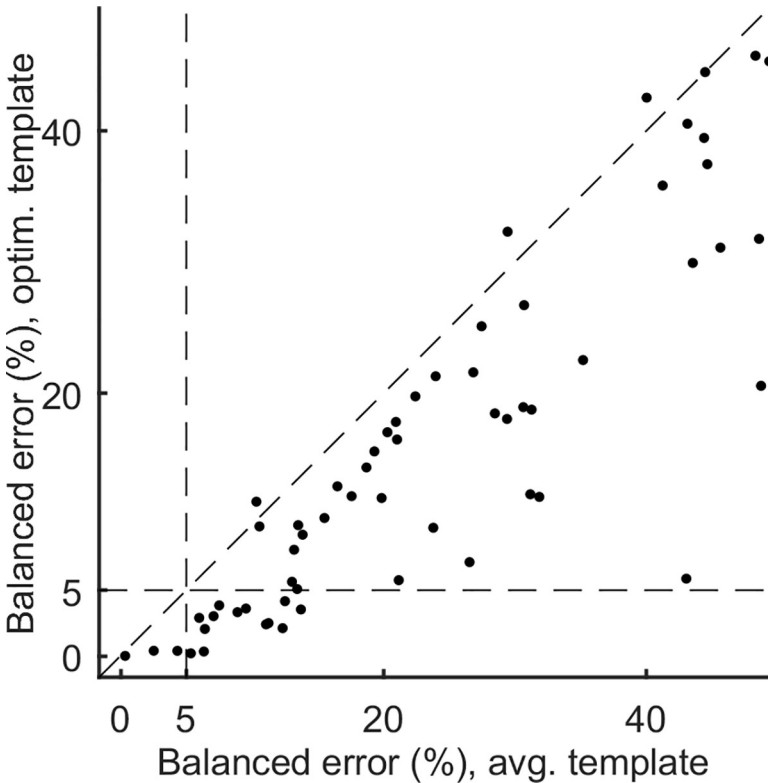

**Fig 5. Pre- vs. post optimization detection performance.** Minimized balanced error for the averaged and optimized templates for all 61 syllables. The horizontal and vertical dotted lines represent 5% balanced error. Optimization increases the number of adequately targetable syllables from 3 to 16.

## Size of training data

The results reported above are based on training data with 25 songs. We also investigated how our algorithm performed with more limited training data. One of the 16 accurately targeted syllables had a total of 12 instances in 25 training set songs and was not evaluated. For the remaining 15 syllables, we divided the 25 training songs into 5 groups of 5 songs and constructed training data sets containing 5,10, 15, and 20 songs by selecting between 1 and 4 groups. By circular cycling through the selected groups, we constructed 5 sets of songs for each training set size. We created and optimized templates from these 20 data sets (4 training set sizes multiplied by 5 groups per size). We also created optimized templates using a single song as training data, randomly selecting 5 individual songs for training. After constructing optimized templates, performance was assessed using the same independent test set of 25 songs for each bird, and these performance values were averaged over the 5 groups with the same training data size.

For the 15 syllables evaluated, targeting accuracy was largely unaffected down to training data size of 10 songs (<0.34% error increase on average; Fig 6). With five songs, balanced error increased 0.67±1.04% (mean ± std, max 3.25%). Even when using a single song for training, balanced error only increased 2.26±1.6% (mean ± std, max 6.5%), and 9 of 15 syllables remained below 5% error.

## Discussion

We have presented a method that improves detection performance for template-based operant conditioning experiments, using syllable detection in songbirds as an example. A common

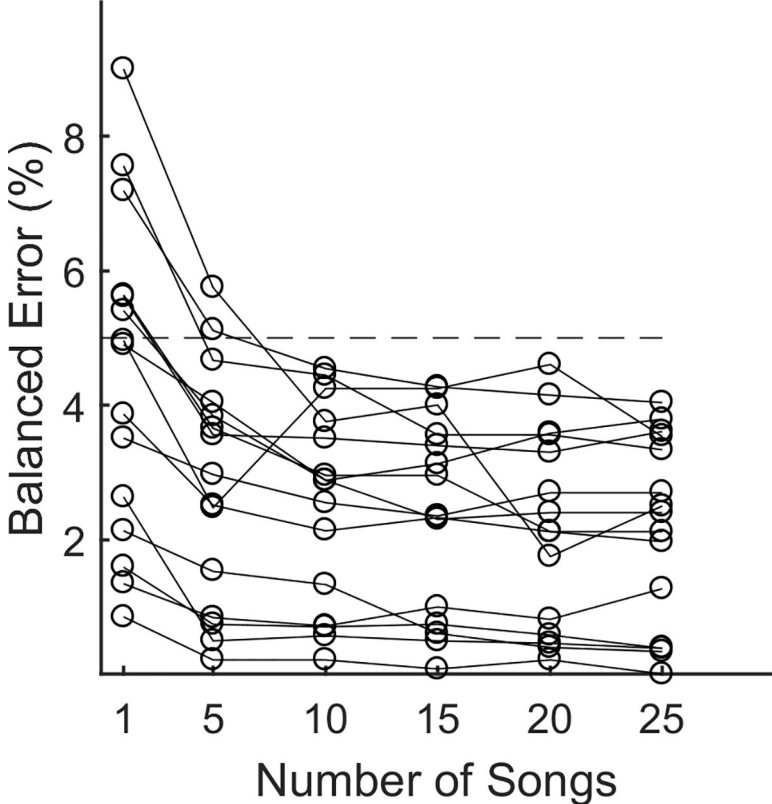

**Fig 6. Performance for small training sets.** Balanced error rates as a function of the number of songs used for training for the 15 syllables that performed below 5% with the full training data. Performance for all templates is evaluated on the same 25 songs in the test set.

approach for providing real-time operant feedback is to create a parameterized template of the behavior of interest, and then provide conditioning feedback whenever behavioral parameters closely match the template. Performance using this approach can be compromised by the presence of distractors, behaviors that closely match the target behavior, but that are not to be conditioned. Our method explicitly views behavioral targeting as a discrimination task and uses gradient descent to minimize both false positive and false negative errors.

We show that the gradient expression for template-based discrimination has a clear and simple geometric interpretation: to increase performance, the template should be moved toward target vectors and away from distractor vectors, with points closer to the decision surface making a greater contribution to the gradient. This procedure can result in a template vector at a significant distance from the initial averaged template, but with an increased threshold for detection. This can result in a decision surface that does a better job of separating target and distractor slices (Fig 2B and 2D, S1 Fig).

For many slices, the optimized template contains values outside the 0–1 boundaries of the normalized spectrum (Fig 2E). This allows the template to emphasize small differences in spectral power near the edges of the 0–1 boundary. For example, suppose that target slices have near zero spectral amplitude at a given frequency, and a distractor has a normalized amplitude of 0.1 there. If the template value at this frequency is 0, the difference in squared distance for the distractor vs. target is 0.01–0 = 0.01. However, if the template value is -0.3, the difference in square distance is 0.16–0.09 = 0.07. In this way, template values outside the 0–1 range can be

used to create a "caricature" of the target spectrum that emphasizes extreme values, making it easier to discriminate targets from distractors.

In songbirds, the main unit of behavior is the syllable, but typical implementations of TF operate by detecting short spectral slices. Syllable segmentation, alignment and normalization introduce complex dependencies between the set of acoustic slices that comprise a given syllable instance, and these dependencies make it difficult to apply standard optimization methods for syllable discrimination. Our method sidesteps these difficulties by first finding an optimal spectral template at the level of individual slices. Calculation of the gradient is straightforward, with each target and distractor slice contributing an independent term to the calculation. After solving this high dimensional optimization, we use a brute force search to select the few remaining parameters (distance threshold, detection criterion) that lead to optimal performance at the level of syllables (Fig 4). In our sample, template optimization increased the number of syllables that could be targeted with fewer than 5% errors from 3 (4.9%) to 16 (26.2%) of 61 syllables. For these syllables, targeting was temporally precise, with a jitter of less than 6 msec (average 3.33 msec).

Our method is based on a widely used implementation of TF first described in [3]. Several other approaches are similar, but perform matching in longer (40–200 msec) spectral windows and use correlation instead of distance as the metric for similarity [9, 22, 28]. In the correlation-based studies, template construction follows the usual method of averaging over a sample of recorded target exemplars. Our discriminant approach can be generalized to these cases and should reduce false positives while retaining a high rate of target detection.

Pearre et al. [29] have also published a syllable detector for zebra finches based on a nonlinear neural network that takes a sliding 30–50 msec window of the spectrum as input. Based on labeled training sets consisting of 1000 songs, they report false negative rates of 1% or less and jitter values around 2 msec. They also report that as few as 200 songs can yield acceptable results [29]. These results appear generally consistent with other studies that have used machine learning based methods [30, 31].

In practice, the effectiveness of TF is often increased by requiring song behavior to match a sequence of two or more spectral templates before providing feedback [11]. This strategy allows the experimenter to exploit temporal regularities of each bird's song to substantially increase the specificity of targeting. Given the complexity of birdsong, the range of combinatorial strategies is large, and varies greatly across birds and across experimental paradigms. For that reason, we did not attempt a systematic investigation of combinatorial targeting. But all such strategies benefit from more precise identification of the elements that comprise the target behavior. Given that our algorithm only concerns the offline selection of optimal spectral templates, it can be used to improve template-based TF with minimal modification to existing software and hardware.

Our method is simple, robust and achieves near asymptotic accuracy when trained on only 5–10 songs (Fig 6). After the song annotation process has been completed, each template optimization takes less than 5 minutes using a standard desktop computer. The intent of many TF experiments is to train birds to alter their vocal production, and in some experiments, templates must be updated regularly to track the changing acoustics of the target syllable. In these situations, ease of training is a critical practical consideration. Moreover, small training sizes allow for easy pre-experiment screening to select birds with targetable syllables appropriate for that experiment. These considerations make our method a practical supplement to common procedures for a wide variety of TF experiments.

## Supporting information

**S1 Fig. Schematic example illustrating the gradient descent algorithm.**
(PDF)

## Acknowledgments

We thank Christopher Moreau for help with the recording hardware. We thank Michael Brainard, Mark Miller, and Daniel Brady for help with EvTAF and related code.

## Author Contributions

**Conceptualization:** Anand S. Kulkarni, Todd W. Troyer.

**Data curation:** Anand S. Kulkarni.

**Formal analysis:** Anand S. Kulkarni, Todd W. Troyer.

**Funding acquisition:** Todd W. Troyer.

**Investigation:** Anand S. Kulkarni, Todd W. Troyer.

**Methodology:** Anand S. Kulkarni, Todd W. Troyer.

**Project administration:** Todd W. Troyer.

**Software:** Anand S. Kulkarni, Todd W. Troyer.

**Supervision:** Todd W. Troyer.

**Validation:** Anand S. Kulkarni, Todd W. Troyer.

**Visualization:** Anand S. Kulkarni, Todd W. Troyer.

**Writing – original draft:** Anand S. Kulkarni, Todd W. Troyer.

**Writing – review & editing:** Anand S. Kulkarni, Todd W. Troyer.

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
