## [Decision Letter · Decision Letter 0]

4 Mar 2020

PONE-D-20-01215

Optimal spectral templates for triggered feedback experiments

PLOS ONE

Dear Dr. Troyer,

Thank you for submitting your manuscript to PLOS ONE. After careful consideration, we feel that it has merit but does not fully meet PLOS ONE’s publication criteria as it currently stands. Therefore, we invite you to submit a revised version of the manuscript that addresses the points raised during the review process.

We would appreciate receiving your revised manuscript by Apr 18 2020 11:59PM. To enhance the reproducibility of your results, we recommend that if applicable you deposit your laboratory protocols in protocols.io, where a protocol can be assigned its own identifier (DOI) such that it can be cited independently in the future. For instructions see: http://journals.plos.org/plosone/s/submission-guidelines#loc-laboratory-protocols

We look forward to receiving your revised manuscript.

Kind regards,

Michael Smotherman

Academic Editor

PLOS ONE

Additional Editor Comments (if provided):

Dear Todd, Both reviewers were very positive. One of the reviewers had a couple of questions that I suspect should be easy to address. The responses to the prior round of reviews appears thoroughly satisfactory, but we couldn't get the same reviewers from last time which requires that we go through the system one more time before a final acceptance can be made.  

Journal Requirements:

All procedures were performed in accordance with established animal care protocols approved by the University of Texas at San Antonio’s Institutional Animal Care and Use Committee (IACUC).

Please amend your current ethics statement to confirm that your named ethics committee specifically approved this study.

For additional information about PLOS ONE submissions requirements for ethics oversight of animal work, please refer to http://journals.plos.org/plosone/s/submission-guidelines#loc-animal-research  

Reviewers' comments:

Reviewer's Responses to Questions

**Comments to the Author**

1. Is the manuscript technically sound, and do the data support the conclusions?

Reviewer #1: Yes

Reviewer #2: Yes

2. Has the statistical analysis been performed appropriately and rigorously? 

Reviewer #1: Yes

Reviewer #2: Yes

3. Have the authors made all data underlying the findings in their manuscript fully available?

Reviewer #1: Yes

Reviewer #2: Yes

4. Is the manuscript presented in an intelligible fashion and written in standard English?

Reviewer #1: Yes

Reviewer #2: Yes

5. Review Comments to the Author

Reviewer #1: Writing and organization of the article:

1. The organization of Results section appears to be confusing. The first subsection of Results “Template optimization using gradient descent” appears to be a mixture of methods and results.

2. In “Calculating averaged templates” in Methods section, the authors explained how the second slice was calculated as a weighted sum of the first (1/15) and the second slice (14/15). Should the second slice be 2/15 or 14/15?

3. In the caption of Fig 1., “and” is redundant in the sentence: For visual clarity, inter-syllable gaps and are not shown.

4. The authors may identify more literature that is similar to [3] and include those in the citation in the sentence: Our work is based on a popular implementation of TF…birdsong field, to demonstrate this specific field is actually popular and thus enhance the significance of this work.

Content and scientific value:

Method:

5. Instead of stretching the shorter instances and compressing longer instances to match the majority - those as long as 16 slices, will averaging only instances of 16 slices enhance or decrease the performance? One may suspect that averaging over different lengths of instances will distort the data, though intuitively only averaging one length of instances may increase the error to detect the same syllable of other lengths.

Discussion:

6. The offline success of this work may be compromised in the online process in actual experiments due to different asparagus, software, e.g., LabVIEW vs. MATLAB, and the requirement to deliver the triggered stimulus as soon as possible, i.e., balancing accuracy and efficiency. What will the detection performance be once the optimal template is applied in an experiment? Could it be the technical limitations after all to decide the hit rate and eventual delay in delivering the stimulus?

7. How long does it take to acquire an optimized template from raw recordings for one syllable? The authors may indicate this in the discussion as well.

Reviewer #2: The manuscript introduces a new method for template optimizations in operant feedback experiments that rely upon on feedback triggered by a specific set of acoustic parameters, such as a note or syllable containing a predetermined set of acoustic features. In these experiments the animals are rewarded for expressing a particular sound/syllable or altering the spectrotemporal properties of the syllables or songs. The challenge is for the automated system to rapidly and accurately detect when the animal has produced the desired output and provide the reward quickly enough to ensure reinforcement. These types of studies are common in songbird research, which is what the paper focuses on, but one can see how this would be a useful improvement for vocal learning studies in other model systems such as marmosets, and might even offer improvements for automated tracking of speech development or efficacy of speech therapies. I’m familiar with this literature and the techniques described here, but cannot offer much in the way of technical suggestions about the methods. In general, however the computational approach is clearly described and presented in a logical way, and would seem to offer a clear benefit over the other methods currently in use. The manuscript is well organized, appears clear of errors and could be published as in. I confirmed that the software package from Github appears to be sufficient and also found the supporting data repository to be helpful.

6. PLOS authors have the option to publish the peer review history of their article (what does this mean?). If published, this will include your full peer review and any attached files.

Reviewer #1: No

Reviewer #2: No

---

## [Author Response · Author response to Decision Letter 0]

1 Apr 2020

We thank the reviewers for their positive comments on our manuscript. Here we address the specific concerns raised by reviewer #1.

1. 'The organization of Results section appears to be confusing. The first subsection of Results “Template optimization using gradient descent” appears to be a mixture of methods and results.'

 It is often uncertain how to divide models or theoretical analysis into “methods” and “results.” We have reorganized the manuscript so that the derivation of our gradient algorithm is included in the Methods section. This section includes some example data to illustrate the method, but all the analysis of performance and group data is presented in Results. We have re-edited the beginning of the Results section to smooth the transition and present our overall approach.

2. 'In “Calculating averaged templates” in Methods section, the authors explained how the second slice was calculated as a weighted sum of the first (1/15) and the second slice (14/15). Should the second slice be 2/15 or 14/15?' 

We have tried several iterations of detailing the calculations that underlie alignment by “linear stretching of time.” At this point we believe that the intuition of linear stretching is sufficiently clear, and have dropped the sentences attempting to detail the math.

3. 'In the caption of Fig 1., “and” is redundant in the sentence: For visual clarity, inter-syllable gaps and are not shown.' 

Done.  

4. 'The authors may identify more literature that is similar to [3] and include those in the citation in the sentence: 

Our work is based on a popular implementation of TF…birdsong field, to demonstrate this specific field is actually popular and thus enhance the significance of this work.'

The references describing the use of triggered feedback in the songbird field were presented in the previous paragraph. We now re-reference the articles that use an averaging approach to constructing templates (most of the references). We retain the single reference [3] where we talk about exactly mimicking the real-time algorithm known as EvTAF, first published in [3].

5. 'Instead of stretching the shorter instances and compressing longer instances to match the majority - those as long as 16 slices, will averaging only instances of 16 slices enhance or decrease the performance? One may suspect that averaging over different lengths of instances will distort the data, though intuitively only averaging one length of instances may increase the error to detect the same syllable of other lengths. '

We share the reviewer’s intuition that restricting the sample size may have the effect of increasing the error. Also, we remind the reviewer that even single spectral slices are averages over some length of the acoustic signal, and that that the windowing inherent in calculating a non-overlapping short time spectrum is not precisely aligned with syllable onsets or offsets. However, to check, we reran our algorithm on a several syllables using only the slices from the modal length syllables as targets. As expected, there was no significant change in the performance of the algorithm.

 6. 'The offline success of this work may be compromised in the online process in actual experiments due to different asparagus, software, e.g., LabVIEW vs. MATLAB, and the requirement to deliver the triggered stimulus as soon as possible, i.e., balancing accuracy and efficiency. What will the detection performance be once the optimal template is applied in an experiment? Could it be the technical limitations after all to decide the hit rate and eventual delay in delivering the stimulus?' 

We remind the reviewer that our algorithm is an offline algorithm whose sole purpose is to provide a better template to use in whatever online algorithm is being used within a real-time experiment. Any technicalities of real time implementation are shared by the original averaged template and the new optimized version. In general, simple online matching algorithms - such as the one we use - have a low computational cost and take less than 1-2 msec on a moderate speed computer. As for performance evaluations, our Matlab algorithm exactly mimics the popular real-time algorithm EvTAF, written in LabView. We have confirmed this match by running our triggering algorithm on data recorded during an EvTAF session.  

7. 'How long does it take to acquire an optimized template from raw recordings for one syllable? The authors may indicate this in the discussion as well.'

It generally takes less than 5 minutes to run the template optimization algorithm after the songs have been annotated. We have included this fact in our discussion.

---

## [Editor Report · Decision Letter 1]

6 Apr 2020

Optimal spectral templates for triggered feedback experiments

PONE-D-20-01215R1

Dear Dr. Troyer,

We are pleased to inform you that your manuscript has been judged scientifically suitable for publication and will be formally accepted for publication once it complies with all outstanding technical requirements.

With kind regards,

Michael Smotherman

Academic Editor

PLOS ONE
---

## [Editor Report · Acceptance letter]

13 Apr 2020

PONE-D-20-01215R1 

Optimal spectral templates for triggered feedback experiments 

Dear Dr. Troyer:

I am pleased to inform you that your manuscript has been deemed suitable for publication in PLOS ONE. Congratulations! Your manuscript is now with our production department. 

With kind regards,

on behalf of

Dr. Michael Smotherman 

Academic Editor

PLOS ONE